# Dynamic infrared aurora on Jupiter

J. D. Nichols [1] ✉, O. R. T. King[1], J. T. Clarke [2], I. de Pater [3], L. N. Fletcher [1], H. Melin [1,4], L. Moore [2], C. Tao [5] & T. K. Yeoman[1]

Auroral emissions are an important diagnostic for a planet's magnetosphere and upper atmosphere. At the outer planets, the characteristics of emission from the triatomic hydrogen ion $H_3^+$ are key to understanding the auroral energy budget. We present James Webb Space Telescope observations of Jupiter's infrared auroral $H_3^+$ emission, exhibiting variability on timescales down to seconds. Together with simultaneous Hubble Space Telescope ultraviolet observations, these results imply an auroral $H_3^+$ lifetime of 150 s, and that $H_3^+$ cannot efficiently radiate heat deposited by bursty auroral pre-cipitation. However, $H_3^+$ radiation is particularly efficient in a dusk active region, which has no significant ultraviolet counterpart. The cause of such emission is unclear. We also present observations of rapid eastward-travelling auroral pulses in the dawn side auroral region and pulsations that propagate rapidly along the Io footprint tail. Together, these observations open a diag-nostic window for the jovian magnetosphere and ionosphere.

Auroral emissions occur when energetic charged particles accelerated in a planet's magnetosphere impinge on the upper atmosphere, depositing large quantities of energy through collisions with atmospheric molecules. The locations and timescales of auroral emissions thus reveal the energetic processes in the magnetosphere that lead to particle acceleration. Auroral energy deposition has been invoked as a possible contributor to the outer planet 'energy crisis' in which these planets upper atmospheres are several hundreds of K hotter than expected on the basis of insolation alone[1]. However, at Jupiter, energetic auroral precipitation also ionises molecular hydrogen $H_2$ to produce $H_2^+$, which rapidly reacts with remaining $H_2$ to produce the trihydrogen cation $H_3^+$. Above the homopause, $H_3^+$ is destroyed by electrons, and can have a substantial lifetime dependent on its dissociative recombination rate[2]. The latter is dependent on the ionospheric electron density and in turn on e.g. auroral precipitation. The $H_3^+$ lifetime is to date poorly constrained, with estimates ranging from 4 to thousands of seconds[3–7], and with sensitivity constraints limiting estimates from ground-based observations to down to around 10–15 min[8]. The lifetime of $H_3^+$ is important since it is thought to be a coolant of the upper atmosphere[3,9–12], strongly radiating ro-vibrational thermal emission in the near-infrared (NIR)[2,8,13–17]. As thermal emission, the $H_3^+$ volume emission rate depends on both the $H_3^+$ density and its

temperature, though in the jovian auroral region changes to the density tend to dominate the observed radiance variation[18]. Under local thermodynamic equilibrium (LTE), each $H_3^+$ molecule radiates $0.5–5\times10^{-20}$ W[19]. Previous observations indicate that overall, $H_3^+$ radiates ~50–60% of the instantaneous gas heating caused by auroral precipitation[12], but the importance of $H_3^+$ in the energy budget cannot be fully determined until the lifetime of auroral $H_3^+$ is well known.

The overall temporal variation of Jupiter's $H_3^+$ emission is, however, not well understood. Jupiter's $H_3^+$ emission has been observed from the ground over many decades[2,8,13–16]. Ground-based observations typically integrate over ~10 min, and are subject to atmospheric seeing of around 0.5″ (or 1,500 km on the planet at opposition), though in 2014 the Subaru Infrared Camera and Spectrograph (ICRS) obtained images with adaptive optics (AO) and achieved 0.2″ pix⁻¹ at 45–100 s time resolution[8], and we discuss these results below. From the vantage point of Jupiter orbit, the Juno Jovian Infrared Auroral Mapper (JIRAM) L-band imager typically images the $H_3^+$ emission at ~100 km pix⁻¹, in a strip 128 × 432 pixels wide hence covering only a small portion of the roughly 50,000 km-wide auroral region in one image[20]. Near perijove, the JIRAM instrument obtains such observations once per 30 s spin of the spacecraft, but for analysis and presentation purposes, pseudo-images built from scanning the imager over the aurora over typically

¹School of Physics and Astronomy, University of Leicester, University Road, Leicester LE1 7RH Leicestershire, UK. ²Center for Space Physics, Boston University, 725 Commonwealth Avenue, Boston 02215 MA, USA. ³Department of Astronomy, University of California, Berkeley, 501 Campbell Hall, Berkeley 94720 CA, USA. ⁴Department of Mathematics, Physics and Electrical Engineering, Northumbria University, Newcastle-upon-Tyne, Tyne and Wear, UK. ⁵Space Environment Laboratory, National Institute of Information and Communications Technology, Koganei, Japan. ✉e-mail: jdn4@le.ac.uk

10–15 min are almost always used[17,21,22]. This technique captures the morphology of features that vary on timescales longer than this but risks misrepresenting the morphology of features that vary significantly faster. A set of five JIRAM images of the Ganymede auroral footprint obtained 30 s apart were shown by[21], though temporal variation of the morphology, other than the expected motion of the footprint, was not discussed. Those authors did make an indirect argument about the chemical lifetime of $H_3^+$ from the physical separation of spots and the speed of the motion of the footprint over the planet. They argued the spots were separated by 60 s of travel time, and they noted that the $H_3^+$ lifetime was therefore possibly less than 1000 s if the precipitating electrons were of sufficient energy to penetrate near to the homopause, where the produced $H_3^+$ would be rapidly destroyed by hydrocarbons. Recently, an analysis of two examples of 30 s temporal resolution JIRAM observations of $H_3^+$ emission in the Io spot has been reported[23]. They concluded that the $H_3^+$ lifetime is likely several hundred seconds, depending on $H_3^+$ number density. However, analysis of temporal variation of the energy input and the resulting $H_3^+$ response has yet to be reported.

An estimated 10–15 min lifetime has been invoked to explain differences in the morphology between the $H_3^+$ and far-ultraviolet (FUV) emission[7,11,12,24]. The latter is 'prompt' Lyman-$\alpha$ and Lyman and Werner band emission from atomic hydrogen H and $H_2$ excited by auroral electron impact[25,26]. Its location and intensity reveals ongoing electron precipitation, and our understanding of Jupiter's magnetosphere-ionosphere system has been significantly advanced by high sensitivity observations of the time-varying morphology of the FUV aurora enabled by e.g. the Space Telescope Imaging Spectrograph (STIS) onboard the Hubble Space Telescope (HST)[27–29]. Equivalent space telescope-based observations have been hitherto unavailable for $H_3^+$. Actual simultaneous FUV and NIR observations are vanishingly rare. There is only one pair of partially-overlapping Earth-based FUV and NIR images available to date[30,31]. The NIR image was obtained using the now-defunct NSFCAM, which yielded substantially blurred morphologies compared with the STIS images, and with only one pair of images, no study of time variability was possible. However, apparently substantial differences in morphology were noted, particularly in the polar region. More recently, concurrent Juno JIRAM and Ultraviolet Spectrograph (UVS) pseudo-images have been compared[11,12]. Several key differences between the FUV and NIR morphologies were highlighted, though we note the above caveat regarding pseudo-image morphology and that the data points compared were rarely simultaneous. Those authors showed that, though the energy dissipated by $H_3^+$ emission is of the same order of magnitude as the gas heating rate, it varies spatially. In particular, it is lowest in bright UV arcs, again potentially indicative of energetic precipitation to an altitude below the homopause. The arrival of James Webb Space Telescope (JWST) offers a similar advancement in our understanding of $H_3^+$ emission to that wrought by HST for the $H_2$ emission.

Here, we present JWST Near Infrared Camera (NIRCam) observations of Jupiter's northern auroral $H_3^+$ emission. These observations provide imaging of the morphology of Jupiter's $H_3^+$ auroral emission at 0.06" pix$^{-1}$ (or ~190 km pix$^{-1}$) spatial and 3 s temporal resolution. The latter is roughly 2 orders of magnitude greater than that typically achieved from the ground, with a negligible noise floor of around 0.5 μW m$^{-2}$ sr$^{-1}$. These high-sensitivity observations evince variability of the $H_3^+$ morphology with timescales down to seconds, and exhibit auroral phenomena including rapidly-propagating pulsations. Furthermore, the NIRCam observations were accompanied by simultaneous HST/STIS images of the $H_2$ emission, providing a unique data set revealing significant time-variable differences between the two wavebands. The comparison enables a determination of the lifetime of auroral $H_3^+$ created following a transient auroral flare, and a measurement of the ionospheric electron number density at auroral latitudes.

## Results
### Morphology overview
Two representative images of the northern jovian auroral $H_3^+$ emission radiance $L$ are shown in Fig. 1. All of the NIRCam images from the programme are listed in Table 1 and shown in Supplementary Movie 1. Many morphological features are broadly similar to equivalent forms in the FUV e.g. ref. 32, though imaged here at the highest temporal resolution ever obtained, and we discuss their temporal behaviour below. The main emission (ME) is a ring of aurora surrounding the magnetic pole, separated at the start of the observation from the high latitude Swirl region on the dawn side by the dark polar region (DPR). In the NIRCam images, this gradually fills in as the observation progresses. Prominent dynamic polar features are a noon active region (NAR), featuring what appears to be a polar auroral filament[33] and dusk active region (DAR). Also evident is the Io footprint (IFP) tail, which maps magnetically to the orbital distance of the moon Io. In this case, the Io footprint itself is just visible off the pre-midnight limb, and the observed IFP tail is ~180° downstream from the satellite.

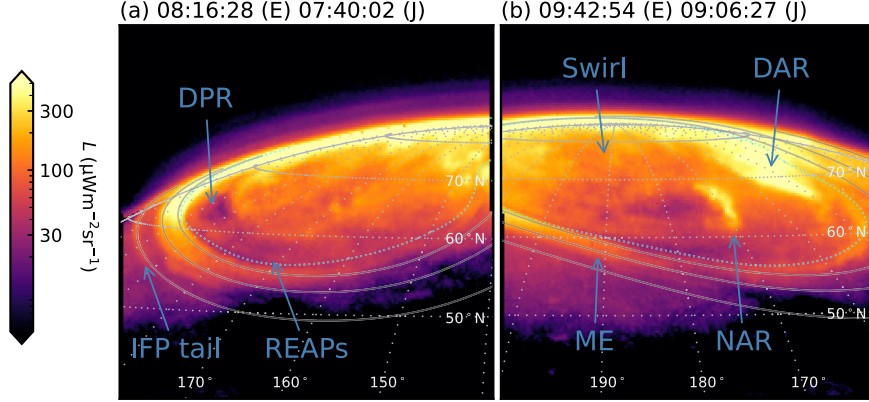

(a) 08:16:28 (E) 07:40:02 (J) (b) 09:42:54 (E) 09:06:27 (J)

**Fig. 1 | Two representative NIRCam images of Jupiter's northern auroral $H_3^+$ emission radiance $L$.** (**a**) Shows the 3 s integration starting at time 08:16:28 UT from observation NIRCam3102, while (**b**) shows the 3 s integration starting at time 09:42:54 UT from observation NIRCam3109. The start time in UT is shown at the top, both at Earth (E) and corrected for one-way light travel time (J). The view is as observed, such that dawn is to the left, dusk to the right, and noon toward the bottom. The dark polar region (DPR), Io footprint (IFP) tail, rapid eastward-travelling auroral pulses (REAPs), Swirl region, dusk active region (DAR), main emission (ME) and noon active region (NAR) are labelled. The planet's limb (grey solid line) and a 10° × 10° graticule (grey dotted lines) at the 1-bar altitude are overlaid, along with solid lines which show the mapped moon footprints (respectively, with increasing latitude Io, Europa and Ganymede)[67] and reference main oval[42] (cyan dotted line) plotted at 500 km above the 1-bar level. Source data are available on Figshare[66].

**Table 1 | Table of observations employed in this study**

| Observatory | Programme ID | Instrument | Observation Start | Observation End | Observation ID |
|---|---|---|---|---|---|
| JWST | 4566 | NIRCam | 2023-12-25T08:14:05 | 2023-12-25T08:24:18 | NIRCam3102 |
| JWST | 4566 | NIRCam | 2023-12-25T08:26:31 | 2023-12-25T08:36:44 | NIRCam3103 |
| JWST | 4566 | NIRCam | 2023-12-25T08:38:52 | 2023-12-25T08:49:05 | NIRCam3104 |
| JWST | 4566 | NIRCam | 2023-12-25T08:51:15 | 2023-12-25T09:01:28 | NIRCam3105 |
| JWST | 4566 | NIRCam | 2023-12-25T09:03:37 | 2023-12-25T09:13:50 | NIRCam3106 |
| JWST | 4566 | NIRCam | 2023-12-25T09:15:59 | 2023-12-25T09:26:12 | NIRCam3107 |
| JWST | 4566 | NIRCam | 2023-12-25T09:28:28 | 2023-12-25T09:38:41 | NIRCam3108 |
| JWST | 4566 | NIRCam | 2023-12-25T09:40:52 | 2023-12-25T09:51:05 | NIRCam3109 |
| JWST | 4566 | NIRCam | 2023-12-25T09:53:11 | 2023-12-25T09:57:35 | NIRCam310A |
| JWST | 4566 | NIRSpec | 2024-01-07T23:47:44 | 2024-01-08T00:05:36 | NIRSpec3101 |
| JWST | 4566 | NIRSpec | 2024-01-08T00:09:32 | 2024-01-08T00:27:17 | NIRSpec3103 |
| JWST | 4566 | NIRSpec | 2024-01-08T00:31:11 | 2024-01-08T00:48:56 | NIRSpec3105 |
| JWST | 4566 | NIRSpec | 2024-01-08T00:53:01 | 2024-01-08T01:10:46 | NIRSpec3107 |
| HST | 17471 | STIS | 2023-12-25T08:57:33 | 2023-12-25T09:32:28 | of8v02dmq |

Times given are the times of observation, i.e. not corrected for one-way light time. The observation ID is a shortened version used here for brevity. To recover the original programmatic observation ID, replace NIRCam with V04566001001P000000000 and NIRSpec with V04566002001P000000000. These data are all available at the MAST archive via doi:10.17909/chb9-zp32 and Figshare[66]. On MAST the JWST observations have observation IDs jw04566-o001_t001_nircam_clear-f335m-sub160 and jw04566-o002_t002_nirspec_g395h-f290lp for the NIRCam and NIRSpec data, respectively.

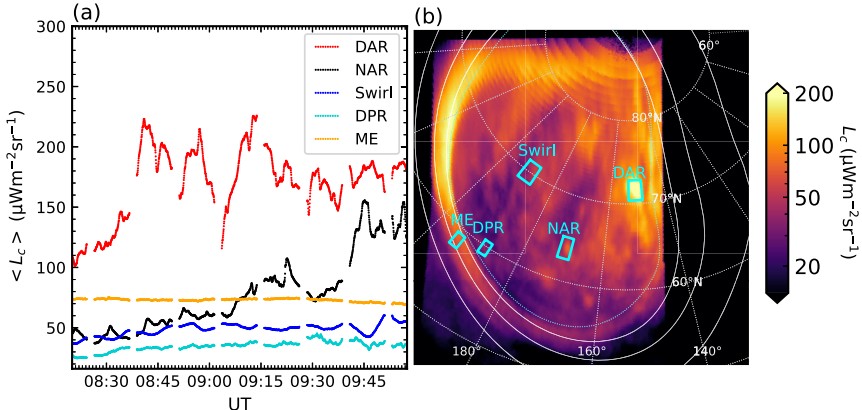

**Fig. 2 | Plot showing the temporal characteristics of different components of the H$_3^+$ emission. a** Shows the mean of the line-of-sight corrected radiance $\langle L_c \rangle$ from regions in the dusk active region (DAR, red), the noon active region (NAR, black), the Swirl region (blue), the dark polar region (DPR, turquoise), and the main emission (ME, orange) versus UT time over the whole observation interval. **b** Shows line-of-sight corrected radiance $L_c$ from a representative image from observation NIRCam3104 projected onto a spheroid 500 km above the 1-bar level and plotted in a polar stereographic projection. The regions from which the $\langle L_c \rangle$ values are computed are shown by the labelled cyan boxes. A 20° × 10° graticule is overlaid in grey, along with solid lines which show the mapped moon footprints (respectively, with increasing latitude Io, Europa and Ganymede)[67] and reference main oval[42] (cyan dotted line). The coordinate system is planetocentric, labelled with System III westward longitude. The central meridian longitude (here 153.78°) is oriented toward the bottom. Source data for (**a**) are provided as a Source Data file, and source data for (**b**) are available on Figshare[66].

## Temporal characteristics

The variability of the H$_3^+$ emission morphology can be appreciated from Supplementary Movie 1. It is also illustrated in Fig. 2, in which we plot (a) the mean intensity throughout the observation from various representative regions illustrated on the projected image shown panel (b). The entire region poleward of the ME is replete with variable (timescale < 10–15 min) transients. The brightest and overall most variable component is the DAR, which exhibits a mean radiance of 170 ± 30 µW m$^{-2}$ sr$^{-1}$ and a maximum gradient of 60 µW m$^{-2}$ sr$^{-1}$ min$^{-1}$. Its radiance can change by significant factors over several minutes. The NAR region, with a mean radiance of 75 ± 31 µW m$^{-2}$ sr$^{-1}$ is somewhat less bright. It is still highly variable with a maximum gradient of 69 µW m$^{-2}$ sr$^{-1}$ min$^{-1}$, which occurs during an 'extinction' such as discussed below. The Swirl region, with mean radiance 47 ± 5 µW m$^{-2}$ sr$^{-1}$, varies gradually on timescales of several minutes, while the dawnside ME is the most stable of

features with 73 ± 1 µW m$^{-2}$ sr$^{-1}$. The DPR exhibits the lowest mean radiance of 34 ± 2 µW m$^{-2}$ sr$^{-1}$, though it exhibits short bursts of emission on timescales of several tens of seconds that will be discussed further below.

As an illustration of the rapidity of the variation of H$_3^+$ emission, we highlight outlier events we term 'extinctions'. In these events, the H$_3^+$ radiance in localised regions suddenly decreased by almost a factor of 2 in ~10 s, amongst otherwise slowly-varying radiance values. An example is shown in Fig. 3, in which the light curve from 76.1° latitude and 159.3° longitude is shown. The radiance at this location increased steadily for the several minutes prior to the event, which began at 09:34:36 UT. Over the next 4 integrations (~12 s) the H$_3^+$ radiance dropped precipitously, reducing in brightness by around 40%. A similar example occurred in the NAR at 09:57 UT. The implications of these extinctions for the H$_3^+$ lifetime is discussed below.

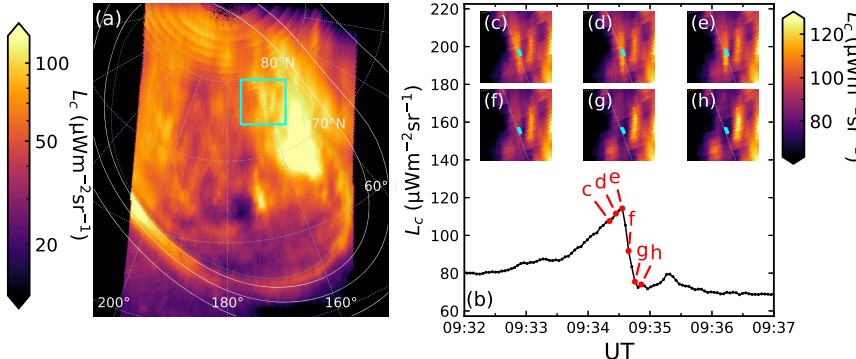

**Fig. 3 | Plot showing an $H_3^+$ extinction event. a** Shows the $H_3^+$ radiance at 09:34:20 UT from observation NIRCam3108, in the same format as Fig. 2 except that 180° System III longitude is oriented toward the bottom. The region of interest is shown by the cyan box, which is about 7100 km in size. **b** Shows the radiance $L_c$ versus UT taken from the location of the cyan squares in (**c–h**) which show the region bound by the box in (**a**) throughout the event as highlighted by labelled red points. Image source data are available on Figshare[66], and (**b**) light curve data are available as a Source Data file.

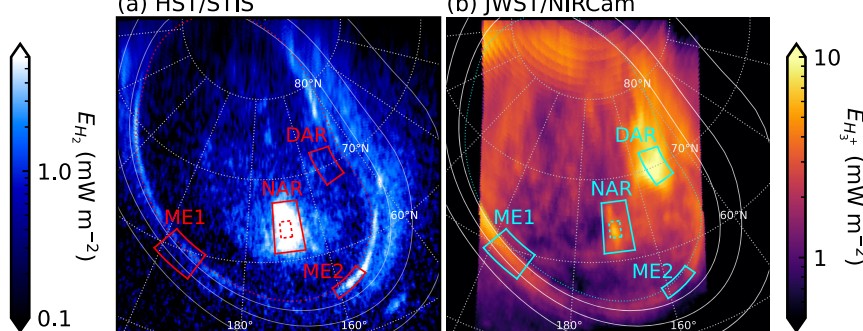

**Fig. 4 | Simultaneous FUV and and NIR observations of Jupiter's auroral emission energy flux. a** Shows the $H_2$ energy flux $E_{H_2}$ from HST observation of8v02dmq, while (**b**) shows the $H_3^+$ energy flux $E_{H_3^+}$ from JWST observation NIRCam3105. Both images were obtained with start time 08:57:33 UT. Images are shown in the same format as Fig. 2, except that the reference main oval is shown with a red dotted line in (**a**). The identical red and cyan solid- (dashed-) line boxes show the regions from which the energy fluxes shown in Fig. 5 (Fig. 6) are derived. Specifically, these are the dusk active region (DAR), noon active region (NAR), and two regions of the main emission (ME1 and ME2) as labelled. Source data are available on Figshare[66].

These observations also exhibit a jovian auroral phenomenon roughly 2° poleward of the ME in the DPR region. Figure 1a highlights a faint band of emission that exhibits rapid eastward-travelling auroral pulses (REAPs) of approximate amplitude ± 3 μW m⁻² sr⁻¹. As discussed, we have produced the keogram shown in Supplementary Fig. 5, which indicates that these pulses propagate eastward along the band at 0.19° s⁻¹. This corresponds to 60 km s⁻¹, or around 20 times the planetary rotation rate. The pulses are of width 30–40 s, separated by on average ~15° in auroral longitude, and Fourier analysis indicates a period of about 1.6 min. We finally here report rapid longitudinal pulsations propagating along the IFP tail at similar speeds to the REAPs. Keograms of two examples are shown in Supplementary Fig. 7, one propagating in either direction. The measured speeds of 0.16° s⁻¹ corresponds to 67 km s⁻¹ or 15 times the planetary rotation rate. We discuss potential origins of these forms below.

**Comparison with FUV images**

We illustrate the relation of the $H_3^+$ and FUV morphologies in Fig. 4, which shows a representative NIRCam image along with a simultaneous HST/STIS image of the FUV emissions. A movie of all simultaneous images is shown in Supplementary Movie 2. For the purposes of comparison, we have converted the units of both sets of images to emission energy flux $E$ in W m⁻² on the planet's spheroid as described in "NIRCam data" of "Methods" section. These images are shown transformed onto polar stereographic projections. The ME and a number of prominent forms in the polar region can be identified in both

wavelengths. However, two striking differences are the intensity of the DPR, which is much darker in the FUV, and the bright $H_3^+$ DAR emission, which is clearly separated from the ME and has little significant counterpart in the FUV.

The comparison is developed in Fig. 5a which shows the mean energy fluxes versus time from the representative regions labelled in Fig. 4. These are the NAR and DAR and two regions of the ME, specifically the narrow, relatively stable dawn-side region (ME1) and a transient and localised bright arc in the post-noon sector (ME2). Prominent features are the relatively large and variable energy fluxes from the $H_3^+$ DAR emission and an $H_2$ emission auroral flare in the NAR, but the relations between the two wavebands are highlighted in panels (b) and (c). Figure 5b shows on the left axis the $H_3^+$ energy flux $E_{H_3^+}$ as a fraction of the precipitating electron energy flux $E_{e^-}$, taken to be ten times the $H_2$ emission energy flux $E_{H_2}$[34]. On the right axis the $H_3^+$ energy flux is shown as a fraction of the instantaneous gas heating rate $E_h$, taken to be $E_{e^-}/2$[11,34]. Thus, values above the grey dotted line represent regions where the $H_3^+$ cooling efficiency exceeds the gas heating rate (DAR and ME1), and vice versa (NAR and ME2). The values range from minima of 0.06 in the NAR and 0.12 in the ME2 bright arc $H_2$ to 2.5 in the DAR, with significant variability observed in most regions; for example the NAR ranges from 0.06 to 1.14. Gérard et al.[11,12] compared contemporaneous UVS and JIRAM intensities. In comparison, they determined overall values of 0.45-0.67, but with an average of 2.4 along a particular radial cut. Our largest value is thus comparable to the latter, but our lowest values are significantly smaller. We also show

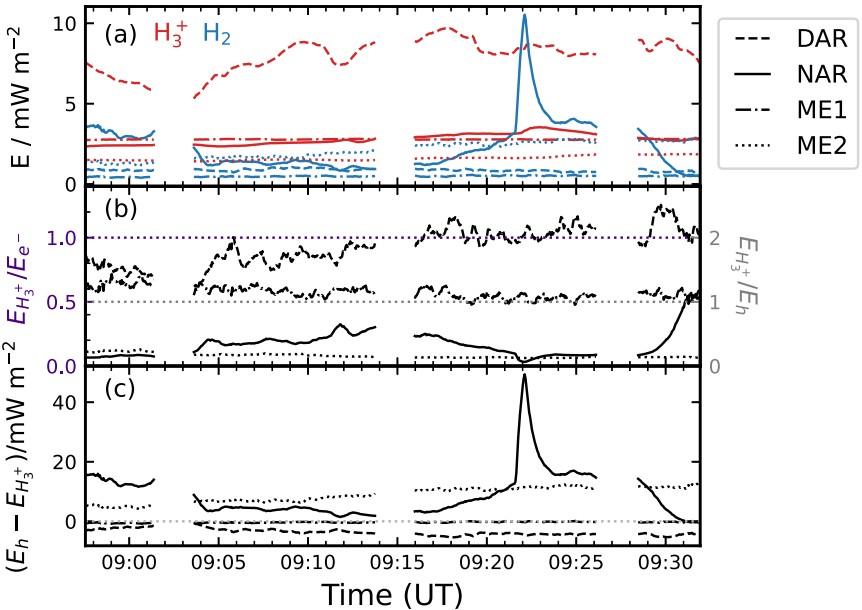

**Fig. 5 | Plot showing simultaneous H₂ and H₃⁺ energy fluxes versus time. a** Shows the mean $H_2$ energy flux $E_{H_2}$ (blue) and mean $H_3^+$ energy flux $E_{H_3^+}$ (red) from the regions as labelled. Specifically, the dashed lines show the results from the dusk active region (DAR), the solid lines those from the noon active region (NAR), the dot-dashed lines those from the dawnside main emission (ME1), and the dotted lines those from the main emission arc (ME2). **b** Shows the $H_3^+$ energy flux $E_{H_3^+}$ as a fraction of the (left axis, indigo labels) precipitating electron energy flux $E_{e^-}$ and (right axis, grey labels) the instantaneous gas heating rate $E_h$, with the same line styles as panel (**a**). The indigo dotted line shows $E_{H_3^+}/E_{e^-} = 1$ and the grey dotted line shows $E_{H_3^+}/E_h = 1$. **c** Similarly shows the residual heating rate $E_h - E_{H_3^+}$. Data are from JWST observations NIRCam3105, NIRCam3106, NIRCam3107 and NIRCam3108, and HST observation of8v02dmq. Source data are available as a Source Data file.

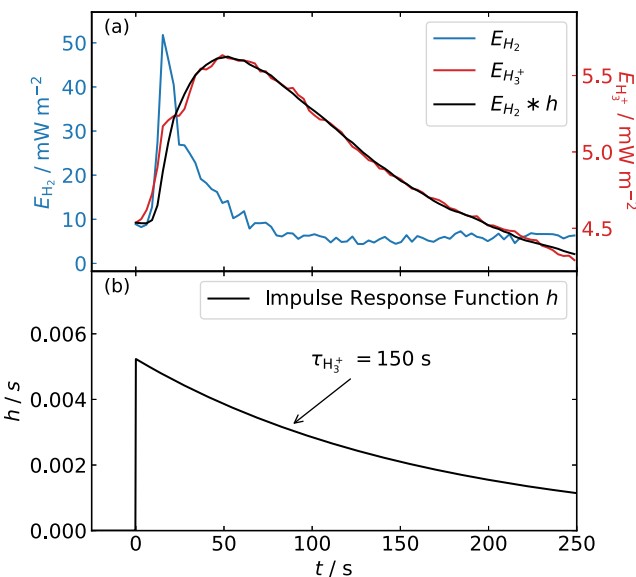

**Fig. 6 | Plot showing the H₃⁺ response to burst of auroral precipitation. a** Shows the $H_2$ energy flux $E_{H_2}$ (blue, left-hand axis) and $H_3^+$ energy flux $E_{H_3^+}$ (red, right-hand axis) from the NAR flare. **b** Shows the best fit impulse response function $h$ labelled with the best fit value of the $H_3^+$ lifetime $\tau_{H_3^+}$. The black line in (**a**) shows the best fit $h$ convolved with $E_{H_2}$. The time axis is in seconds from the start of the flare. Source data are available as a Source Data file.

that the values observed at any one point exhibit significant temporal variability. The overall significance of the $H_3^+$ cooling efficiencies can be appreciated by considering the residual heating rate $E_h - E_{H_3^+}$ shown in Fig. 5c. The NAR and ME2 both exhibit mean net heating rates of 8.9 mW m⁻², while the DAR and ME1 have values of -3.7 and -0.4 mW m⁻², respectively, i.e. net cooling. It is notable that the temporally stable ME1 region exhibits such a small value, indicating the $H_3^+$ cooling essentially balances the electron precipitation-induced gas

heating rate in this region. The other regions that deviate from this balance are characterised by transient phenomena such as flares, localised bright arcs and the DAR. In particular, the flare observed in the NAR shows that such events are particularly significant sources of heating, with the net heating rate peaking at 49 mW m⁻². Integrated over its area and ~2 min duration, this flare deposited 55 TJ of heating, with $H_3^+$ emitting 0.9 TJ integrated over 10 min following the spike start. This illustrates the inefficiency of the $H_3^+$ thermostat in such events.

This flare further offers an opportunity to measure the $H_3^+$ lifetime by considering the H₂ (blue) and H₃⁺ (red) light-curves shown in Fig. 6a. Note, for this particular analysis we have extracted the HST data with 3 s integration to match the JWST observations, and we take the mean from the region bounded by the dashed boxes in Fig. 4. This region is focused on the centre of the flare in both images. Comparison of the forms of the light-curves illustrates vividly the broadened temporal response of the $H_3^+$ emission compared with the electron energy input. To determine the $H_3^+$ lifetime, we have created an impulse response function $h(t)$, which represents how the $H_3^+$ energy flux would evolve in response to a brief pulse of unit magnitude at time $t = 0$, of the form

$$h(t) = \begin{cases} 0, & \text{if } t < 0 \\ A \exp^{-t/\tau_{H_3^+}}, & \text{if } t \geq 0 \end{cases} \quad (1)$$

indicating instantaneous rise to amplitude $A$ followed by exponential decay governed by timescale $\tau_{H_3^+}$. This form is expected since the $H_3^+$ production rate is understood to be fast[2], while the destruction rate is to be determined. Convolution of $h$ with the input $E_{H_2}$ (minus the baseline value at the start of the FUV spike, 9 mW m⁻², to normalise the flux signal) while respecting causality, i.e.

$$E_{H_2} * h(t) = \int_0^t E_{H_2}(t') h(t - t') dt' \quad (2)$$

then yields the expected $E_{H_3^+}$ response for the given parameters $A$ and $\tau_{H_3^+}$. These parameters were optimised by fitting $E_{H_2}*h(t)$ to $E_{H_3^+}$[35,36],

and the uncertainty determined by Monte Carlo Markov Chain (MCMC) parameter estimation[37]. This yields $\tau_{H_3^+} = 150 \pm 4$ s and $A = (5.23 \pm 0.06) \times 10^{-3}$ s$^{-1}$. The best fit $E_{H_2} * h(t)$ is shown by the black line in Fig. 6a. The quality of the fit indicates the chosen impulse response function represents the behaviour of $H_3^+$ very well.

## Discussion

These results show that, overall, Jupiter's $H_3^+$ morphology is more variable than inferred from previous observations. In particular, by considering the light-curves of both $H_2$ and $H_3^+$ emission in an auroral flare we have measured the $H_3^+$ lifetime $\tau_{H_3^+}$ to be $150 \pm 4$ s. However, the extinction event shown in Fig. 3 indicates that the $H_3^+$ lifetime can take even shorter values, of order 10 s. Assuming the latter is driven by $H_3^+$ number density, an estimate of the $H_3^+$ lifetime $\tau_{H_3^+}$ then follows from assuming exponential decay, such that $\tau_{H_3^+} = 12 / \ln(114/75) = 29$ s. However, given that the value of 150 s is uniquely constrained by both FUV and NIR observations and the extinctions are outlier events, we regard $\tau_{H_3^+} = 150$ s as the primary result but note that it likely takes values down to 29 s. Putting this in context of previous observations, Watanabe et al.[8] reported observations of Jupiter's auroral $H_3^+$ emission obtained in 2014 with Subaru ICRS. They took NIR images with AO and achieved 0.2″/pix at 45–110 s time resolution. They reported pulsations in a polar form with a period of around 8 min, but the fastest relative drop they observed was from 20.219–18.053 µW m$^{-2}$ sr$^{-1}$ over a pair of observations near 7:18 UT separated by 109 s. A similar gradient was observed near 6:58 UT. Their narrowband filter passband of 0.022 µm differs from the F335M filter, such that direct comparison of absolute values with our results is not appropriate. However, we note that this represents a drop in radiance of 10.7% in 109.2 s. Watanabe et al.[8] did not explicitly estimate $\tau_{H_3^+}$ from their results, but Miller et al.[2] cited $\tau_{H_3^+} = 110$ s, as the resulting estimate of the lifetime. In light of our results, this appears to underestimate the implied lifetime. For comparison, in the extinction shown in Fig. 3, the radiance falls by ~10% by the first integration, i.e. in 3 s. Applying our method to Watanabe et al.[8]'s results instead yields $\tau_{H_3^+} = 109.2 / \ln(20.219/18.053) = 964$ s (i.e. 16 min). For consistency, we take these values for comparison with our results. As discussed above, Mura et al.[23] recently reported $H_3^+$ lifetimes based on modelling of 30 s resolution observations of the $H_3^+$ emission in the Io spot. They concluded that the $H_3^+$ lifetime is inversely proportional to the number density, and their inferred values ranged from ~60–600 s. Those observations were not analysed in conjunction with simultaneous FUV observations. Our observations at 3 s resolution and constrained by simultaneous FUV observations indicate that exponential decay fits the observed radiance profile satisfactorily. Having said that, an $H_3^+$ lifetime that varies with local conditions (if not directly related to radiance) is borne out by our observations, and ionospheric modelling work constrained by these data will shed further light on the behaviour of $H_3^+$ in response to an auroral flare.

Two important implications follow from the determination of the $H_3^+$ lifetime: first, this provides the auroral ionospheric electron number density at the altitude of peak $H_3^+$ emission (which we take to be around 500 km), which can be estimated assuming the variation in radiance can be ascribed primarily to a change in the $H_3^+$ column density or dominated by the $H_3^+$ density at the peak. Then, the auroral ionosphere electron number density $n_e$ can be estimated via $\tau_{H_3^+} = (\kappa n_e)^{-1}$, where $\kappa = 1.15 \times 10^{-7}$ cm$^3$ s$^{-1}$ is the dissociative recombination rate coefficient[2]. This yields $n_e = 5.8 \times 10^4$ cm$^{-3}$ for $\tau_{H_3^+} = 150$ s, and $n_e = 3.1 \times 10^5$ cm$^{-3}$ for $\tau_{H_3^+} = 29$ s. The former is highly consistent with the mean value of ~$6 \times 10^4$ cm$^{-3}$ in the highest northern latitude bin of the maximum electron numbers densities determined from Galileo radio occultations. However, we note that those observations contain essentially no data below 1000 km altitude and were obtained on the terminator, rather than near noon in our case[38]. The latter is likely unrepresentative of the typical number density but is consistent

with the highest values observed at lower latitudes[38]. It is worth also comparing our results with the values in the auroral regions of up to $10^{10}$ cm$^{-3}$, determined by Hodges et al.[39] from Juno microwave radiometer measurements. Those values are clearly significantly different, though may represent upper limits to local density values. Future ionospheric modelling work and comparison of JWST data with e.g. Juno radio occultation measurements will further constrain the electron number densities.

Second, this short 150 s lifetime indicates that $H_3^+$ cannot effectively radiate energy deposited by bursty precipitation since the $H_3^+$ molecules do not survive for a significant length of time. The effect of the $H_3^+$ lifetime $\tau_{H_3^+}$ on the total energy radiated following an impulsive burst of precipitation can be simply illustrated if we again assume that the column density, and hence radiance, is primarily controlled by $H_3^+$ number density that decreases exponentially with time. The latter is suggested by the success of the impulse response function in reproducing the observed $H_3^+$ energy flux following the NAR flare. Then, since $\int_0^\infty \exp^{-t/\tau_{H_3^+}} \, dt = \tau_{H_3^+}$, the total energy emitted is proportional to the lifetime, and a lifetime of 150 s indicates that $H_3^+$ only radiates ~15% of the energy from a precipitation burst than if its lifetime were the 16 min implied by the Watanabe et al.[8] observations. Indeed, $H_3^+$ radiated only 2% of the 55 TJ of thermal energy deposited by electron precipitating in the NAR flare. This flare contrasts with the heating event studied by Melin et al.[9] in that the precipitating electron energy flux is a factor of 8 higher, and the $H_3^+$ cooling a factor of 2 lower, implying that either hydrocarbon cooling or downward conduction[40] must account for the excess heat to a greater degree than found for that event. We note this argument only applies to variable precipitation and not to e.g. the dawnside ME, which is remarkably steady, and for which $H_3^+$ emission essentially balanced the gas heating rate. It is also partially offset by the regions, e.g. the DAR, where the $H_3^+$ energy flux is greater than the gas heating rate. However, in that case the magnitude of the difference was smaller than the cases for which the gas heating rate was larger. The latter corresponded to brighter, transient or variable FUV auroral forms for which, as pointed out by Gérard et al.[11], the precipitating electron energies are likely energetic enough to precipitate below the homopause, where $H_3^+$ is rapidly destroyed.

The bright, rapidly varying DAR emission poses a challenge for our understanding of the jovian auroral ionosphere. The emitted energy flux is greater than the precipitating energy flux, indicating the source is not simply precipitation. Other sources of heat include ion drag and Joule heating. However, as discussed by Watanabe et al.[8], timescales for temperature changes ($10^3$-$10^4$ s K$^{-1}$) and transport and diffusion ($10^4$-$10^5$ s) cannot account for $H_3^+$ emission variability on even 10 min timescales. Hence, the rapid changes evident in the DAR are likely due to creation and destruction of $H_3^+$. This is supported by the JWST Near-infrared Spectrograph (NIRSpec) observations shown in Fig. 7. Although these data were obtained a few weeks after the NIR-Cam observations, they exhibit a similar bright form in a similar region to the NIRCam DAR. This feature appears to be associated with enhanced column density, rather than temperature, since there is no corresponding enhancement in the temperature in this region. This behaviour is consistent with previous observations[18,22]. Given the evident short lifetime of $H_3^+$, and the variability of the emissions during the observation interval, it is unlikely that the form is an afterglow following a burst of precipitation prior to the observations. The region poleward of the main emission on the dusk side often exhibits bright, variable patches and arcs in the FUV[41,42]. In the NIR, relatively bright patches have been observed from ground-based observations on the dusk side[15,18,43], though with lower spatial resolution than in the images presented here. It is unclear whether these are related to the DAR emission, which is clearly an isolated form separated from the main emission. As discussed above, Juno JIRAM has observed forms (arcs and patches) poleward of or branching off the main emission[11,12,22], and

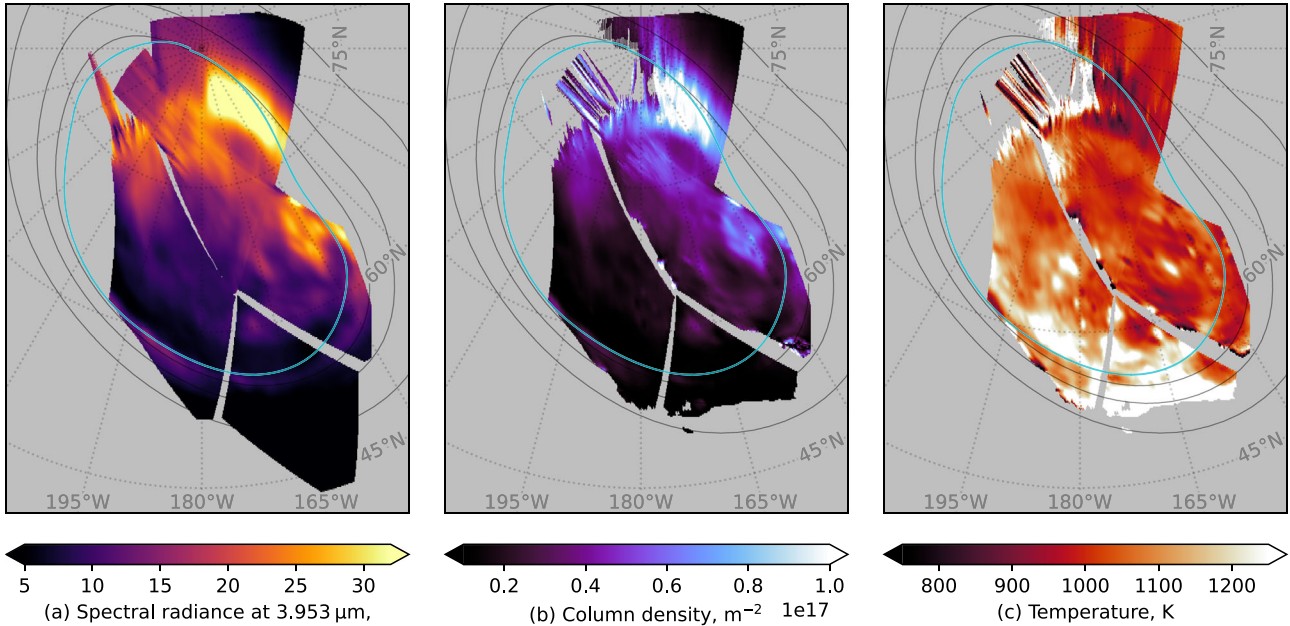

**Fig. 7 | Plots showing NIRSpec observations of the northern $H_3^+$ emission.**
**a** Shows values of the 3.953 $\mu$ spectral radiance averaged over the observation for each location. **b** Shows the inferred column density and (**c**) shows the temperature. Grey indicates no data or no fit convergence. A 15° × 15° graticule is overlaid in grey, and satellite and reference main oval contours are plotted as for Fig. 2. The projection coordinate system is as for Fig. 2 except that 180° System III longitude is oriented toward the bottom. Data are from all NIRSpec observations in the programme. Source data are available as a Source Data file.

the $H_2$ and $H_3^+$ intensities have been compared. The poleward forms observed by Gérard et al.[12] were present in regions of broadly elevated FUV intensity, even if the locations of the $H_2$ and $H_3^+$ maxima did not precisely coincide. Note also those forms were observed in pseudo-images built over 10–15 min. In contrast, the images obtained here show the bright DAR emission to be a highly variable form, separated from the main emission and in a region of generally low FUV intensity.

In order to examine the cause of the DAR emissions, we have estimated the precipitating electron population necessary to drive the mean FUV and $H_3^+$ emission observed in the DAR region using the Tao et al.[6] model. This model estimates UV and IR volume emission rates arising from ionisation and excitation in the $H_2$ atmosphere caused by auroral electron precipitation. The model includes UV absorption by hydrocarbons, ion chemistry, and $H_3^+$ non-LTE effects. Before discussing the model results, we first note that the means of the region contained with the DAR box shown in Fig. 2b during the simultaneous JWST and HST observation interval are 175 $\mu$W m$^{-2}$ sr$^{-1}$ for the $H_3^+$ radiance and 170 kR for the FUV intensity. Modelled values are presented in Fig. 8, which shows (a) FUV intensity and (b) $H_3^+$ radiance, varying with both precipitating electron energy $E$ and upward ionospheric field-aligned current density $j_{\|i}$. The 170 kR FUV intensity contour is also plotted in panel (b) to highlight the region that broadly matches the observations, as shown in the region highlighted by the green ellipse in panel (b). This region corresponds to ~1 keV electrons and 4 $\mu$A m$^{-2}$ current density. However, this combination of parameters is problematic, since high latitude jovian electrons with this energy cannot carry 4 $\mu$A m$^{-2}$ current density. This energy is similar to the value for the hot electrons outside the equatorial current sheet typically taken as the source population for jovian auroral processes, i.e. 2.5 keV and 0.01 cm$^{-3}$[44]. This unaccelerated electron population is only capable of carrying a maximum of around 0.01 $\mu$A m$^{-2}$[45], and larger currents thus require field-aligned acceleration, which increases the energy of precipitating electrons. With this source population, the Cowley[46] relativistic current-voltage relation shows a minimum accelerating potential of ~30 kV is

required to drive a current of around 1 $\mu$A m$^{-2}$. Observations of electrons above and poleward of the jovian auroral region with the Juno JADE and JEDI instruments have indeed indicated that poleward of the ME the characteristic energy is around 100 keV and very little energy flux carried by electrons with energy <30 keV[47]. Hence, the origin of this combination of FUV and $H_3^+$ emission is unclear, and points to a fundamental gap in our understanding of the jovian ionosphere and the production of $H_3^+$. Such an uncertainty clearly impacts efforts to predict the $H_3^+$ luminosities expected from e.g. brown dwarfs based on electron precipitation[48]. Future work more closely examining the electron precipitation above the DAR in Juno data is recommended.

Likewise, the REAPs pose challenges for jovian magnetospheric physics. They map to 40–60 $R_J$, and at this distance would correspond to a linear speed of ~10,000 km s$^{-1}$. We note the coincidence between the 1.6 min period and the gyroperiod of a singly-charged oxygen ion O$^+$ or a doubly-charged sulphur ion S$^{++}$ in a 10 nT field, and indeed Petkaki and Dougherty[49] observed electromagnetic ion cyclotron waves with this period in the equatorial morning magnetosphere at these distances in Ulysses magnetometer data. These could weakly scatter electrons into the loss cone to produce pulsing $H_3^+$ emission, but the origin of the perpendicular propagation is unclear; it is likely a perturbation phase speed. Considering the IFP tail pulsations, we note that the auroral longitude speed of 0.16° s$^{-1}$ corresponds to 15 times the planetary rotation rate, which at 5.9 $R_J$ is equivalent to ~1100 km/s. According to[50], this Alfvén speed is met roughly 90 $R_{Io}$ = 2.3 $R_J$ up the field line from the equator, which broadly corresponds to the extent of the Io plasma torus[51]. We suggest that this perturbation could therefore represent fast mode waves propagating along the edge of the IPT, that must couple with e.g. a shear Alfvén mode to produce the aurora observed. Future modelling work beyond the scope of the present paper is required to understand the nature of both of these auroral perturbations. Overall, these observations, though spanning only a brief period, demonstrate the value of JWST/NIRCam for the study of the auroras of the outer planets.

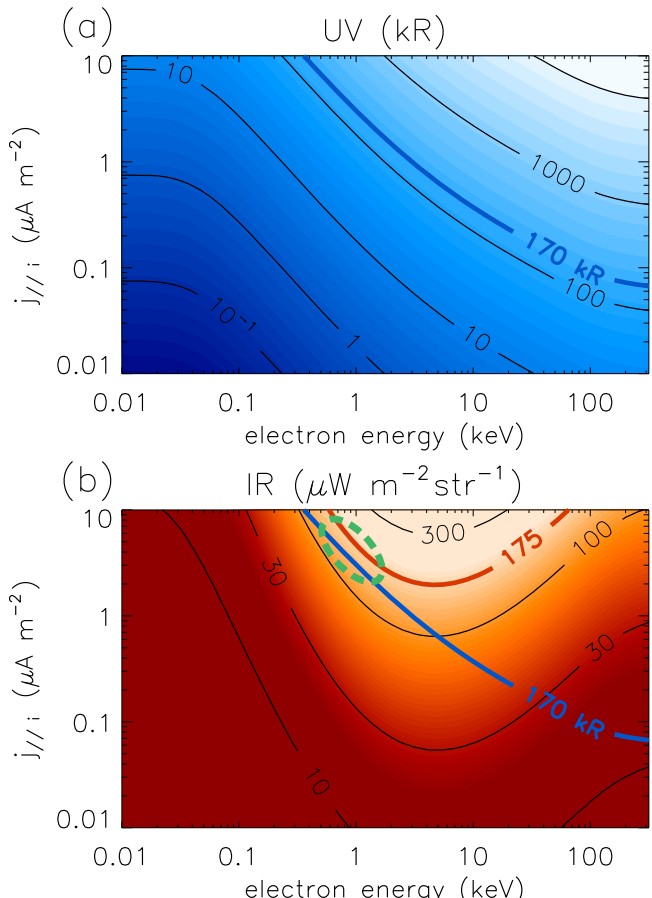

**Fig. 8 | Modelled precipitating electron parameters required for DAR FUV and NIR combination. a** Shows the total unabsorbed $H_2$ emission intensity over 70–180 nm and (**b**) shows the $H_3^+$ radiance in the F335M window, each versus electron energy and field-aligned current (FAC) density $j_{||i}$ from the Tao et al.[6] model. The mean values from the DAR region are shown using thick lines, and this FUV line labelled 170 kR is also shown in (**b**) for comparison with the $H_3^+$ emission. The region broadly matching the DAR observations is shown by the green dashed ellipse in (**b**). Source data are available as a Source Data file.

## Methods

### NIRCam data

These observations were obtained as part of JWST programme 4566 on 25 December 2023, using the long wavelength channel of NIRCam with the CLEAR pupil aperture stop and F335M filter. The latter admits light between 3.18 and 3.54 μm. This wavelength range covers several intense ro-vibrational R-branch $H_3^+$ spectral lines as well as coinciding with a strong methane absorption band across 3.2–4.0 μm, such that it provides excellent contrast against the background of reflected sunlight and thermal emission from the relatively dark planetary disc. The observation times are given in Table 1. The accompanying NIRSpec observations discussed below indicate, by comparison with fitted model $H_3^+$ spectra[52,53], that the F335M filter admits around 25% of the total $H_3^+$ emission. Some methane fluorescence lines and reflected sunlight from the deeper atmosphere are still present, and we discuss their contribution to the overall radiance below. Observations were obtained using the 10 × 10" SUB160 subarray on the NIRCam detector, which covered the majority of the auroral region, with an effective integration time of ~3 s and total exposure time of 4698 s. The field of view was centred on 67° planetographic latitude, 175° System III westward longitude, thus tracking the auroral region as the planet rotated. No pixels were saturated. The raw uncal data files have been processed with calwebb (version 1.14.0)[54] stages 1 and 2 to produce the calints

product containing the calibrated integrations in spectral radiance units of MJy sr$^{-1}$. For presentation we have converted values to radiance in μW m$^{-2}$ sr$^{-1}$ using the filter passband width of 0.347 μm. Unsurprisingly, the pipeline detected and flagged count rate jumps due to cosmic rays and regions where the auroral radiance changed rapidly, but no pixels were flagged as Do Not Use beyond the usual bad pixels on the detector. We cosmetically corrected bad pixels using bilinear interpolation, and performed a correction for effects due to charge bleeding onto the SUB160 subarray from external pixels not reset every integration. In this mode, pixel regions adjacent to each edge of the subarray are reset every 4 integrations. Where the auroras were bright, those external pixels saturated, and the charge bled into the used subarray, causing a flickering of the nearest pixels. We corrected this by transforming the time series of each pixel into the frequency domain, interpolating over spikes with frequencies of 0.5 and 0.25 per integration, and transforming back to the time domain. Though only around 3 rows of pixels near the edge were substantially affected, we performed this fix for each pixel to be sure there were no spurious variations remaining.

We then rotated each image such that the planet's north pole was oriented upward, and then scaled the planet to a standard distance of 4.2 astronomical units (au) to facilitate comparison with any other observations. We determined the observation navigation by stitching together the first and last image of the visit, and then using the limb and the Io footprint tail to infer the planet's centre pixel. We then obtained the (assumed constant) offset from the expected centre pixel position given by the Navigational and Ancillary Information Facility (NAIF) Spacecraft, Planet, Instrument, C-matrix, Events (SPICE) toolkit[55]. We used this to compute the centre positions for each integration from the reference pixel right ascension and declination values given in the uncalibrated data files. The SPICE kernels used are naif0010.tls, pck00010.tpc, de421.bsp, jup230l.bsp, and jwst_rec.bsp. Following this step, the computed limb and planet positions broadly agreed throughout each exposure apart from a few single-integration jitters, in which the centre pixel coordinates for these integrations differed from their neighbours by several pixels causing the planet to jump relative to the expected location. This was corrected by fitting 3rd-order polynomials to the planet centre pixel coordinates, and then redefining the centre coordinates to the fitted values. Most coordinates were essentially unchanged by this procedure but the outliers were corrected and the jumps disappeared. The images exhibited low-level (<1%) random flickering, indicating some residual noise, which was corrected by background subtraction. We subtracted the background separately for the planet's disc and the sky. For the sky, we calculated and subtracted the mean of a 4-pixel wide strip across the top of the image (i.e in the sky). The background on the disc is temporally stable on the timescale of one observation, but spatially non-uniform. It comprises both methane emission from low altitude features and non-auroral $H_3^+$. In the absence of detailed models of the background, we have computed and subtracted a constant value given by the mean of a 4 × 4 pixel box centred on 166° System III longitude, 47° latitude. This location was chosen to set to near-zero much of the low latitude disc, and leave a small residual westward of 190° longitude.

We then projected the emission to a spheroid with an altitude 500 km above the 1-bar level, an altitude used previously for $H_3^+$ observations[e.g. 17], preserving intensity. For the projected images, we have applied a simple line-of-sight correction to the radiance values $L$ for this strongly limb-brightened $H_3^+$ emission by multiplication by the cosine of the observation angle to give corrected radiances $L_c$. While this is a standard correction[e.g. 30,56,57], emission close to the limb may not be accurately corrected. For comparison with the HST images, we have converted the unprojected, uncorrected $H_3^+$ radiance (in W m$^{-2}$ sr$^{-1}$) images into total $H_3^+$ power in W pix$^{-1}$ assuming isotropic emission and projected the images onto the planet, this time conserving energy

rather than intensity to produce W per projection grid cell. Dividing by the area per projection grid cell then yields energy flux $E_{H_3^+}$ in W m$^{-2}$ on the planet.

## NIRSpec data

As stated above, as well as $H_3^+$ lines, the NIRCam F335M passband admits light from methane (CH$_4$) fluorescence near 3.3 μm and variable reflected sunlight from the upper troposphere and lower stratosphere, from within a deep methane absorption band. While this study focuses on the NIRCam observations obtained during JWST programme 4566, we also obtained NIRSpec Integral Field Unit (IFU) observations on 8 Jan 2024, which can be used to estimate the non-$H_3^+$ contribution to the radiance observed with NIRCam. NIRSpec IFU spectroscopy of the auroral region over 2.9–5.3 μm was obtained with the G395H/F290LP disperser/filter setup. As with the NIRCam observations, the integrations, here 32 s in length, are the scientific product of primary interest. The raw uncal data files were first processed through calwebb stage 1, and the resulting files split into integrations. Each integration was then processed through stages 2 and 3. For the latter, the 'emsm' weighting was used, as it was found to minimise artefacts from the cube-building step. The generated spectral cubes were then re-merged to create 4D arrays. The observation navigation was again determined by reference to the limb, and the field of view was projected onto a spheroid 500 km above the 1-bar level, as for the NIRCam observations.

Supplementary Fig. 1 shows the relevant portion of representative NIRSpec spectra from (a) the Swirl region and (b) the DAR region, along with an indication of the F335M passband and fits to the spectrum using an MCMC fitting routine, based on the h3ppy code[52]. Following[58], the spectral lines were fitted using the wavelength ranges shown in yellow. The background comprises both an order 10 polynomial fit to the continuum and fitted methane lines. The $H_3^+$ emission is evident as the red excess above the cyan background. The R-branch $H_3^+$ lines are clearly evident, along with the solar reflectance increasing toward the short wavelength end of the passband. The methane contribution in the Swirl region is larger than for the DAR region, as we also show below.

Supplementary Fig. 2 shows light curves indicating how the proportion of $H_3^+$ radiance changes at two representative locations; (a) in the swirl region, and (b) near the dusk active region. While the overall level of non-$H_3^+$ emission is different in the two cases, it does not vary strongly. Instead, the variability of the total radiance in each case is driven strongly by the $H_3^+$ component, and the fraction of $H_3^+$ increases as the total radiance increases. Quantitatively, the Pearson's correlation coefficient between the $H_3^+$ and the total radiance is >0.99. This justifies ascribing the rapidly varying features observed in the NIRCam data to $H_3^+$, however we acknowledge that a small proportion (around 10 %) is potentially CH$_4$ fluorescence excited by auroral precipitation.

In Supplementary Fig. 3, we show projected maps of the total radiance convolved with the F335M throughput, the inferred $H_3^+$ radiance in the F335M passband, and the $H_3^+$ contribution to the total radiance averaged over every location observed by NIRSpec. It is evident that the total fraction of $H_3^+$ intensity varies throughout the auroral region, and we discuss below modelling and subtracting a constant methane component.

## Methane fluorescence subtraction

As shown in Supplementary Fig. 3, the $H_3^+$ contribution in the swirl region falls to around 30–40%, with the remainder originating from methane. This is consistent with previous observations of a region of methane emission contained within the auroral oval[59]. This region of methane emission is fixed in SIII longitude, and, as shown in Supplementary Fig. 2, relatively stable in radiance. We have therefore created a simple model of the CH$_4$ emission in this region and subtracted the values from the NIRCam radiances. The CH$_4$ emission radiance is

modelled with a two-dimensional Gaussian of the form

$$L_{CH_4} = A \exp\left(-0.5\left(\frac{x^2}{3.7^2} + \frac{y^2}{7.2^2}\right)\right) \tag{3}$$

centred on 67° latitude, 175° System III longitude, where $x$ and $y$ are measured in degrees and $y$ is oriented along a bearing 33° east of north. The same functional form is used for both $L_c$ and $E_{H_3^+}$, with amplitude $A$ equal to 54 μW m$^{-2}$ sr$^{-1}$ and 2.75 mW m$^{-2}$, respectively. We show this model background in Supplementary Fig. 4b, along with (a) the original image and (c) the residual following subtraction. A cut along the latitude 70° N is also shown in panel (d), which illustrates the excess emission in the swirl region is effectively removed. Remaining small scale features are transients. We should emphasise that, while removing this background is, in our view, an appropriate processing step, none of the key results in this study are dependent on the absolute value of the radiance in the swirl region since we concentrate on time variability.

## STIS data

The HST/STIS observations were obtained in programme 17471 using the F25SRF2 filter, which admits light from the H$_2$ Lyman and Werner bands, while rejecting H Lyman-$\alpha$. Data were obtained in time-tag mode, and images were extracted with a 30 s integration at the starting time of each NIRCam integration (which recall had a 3 s cadence). Images were reduced using the Leicester implementation of the Boston University pipeline[60], which has been employed many times previously e.g. refs. 61–63. Specifically, the calibration steps are: dark count subtraction; flat field response correction; geometric distortion correction; rotation to place the planet north upward, and rescaling image size to 4.2 au; conversion of count rates into brightness units of kR per pixel of total un-absorbed H$_2$ emission over 70–180 nm using the conversion factor 4523, corresponding to a colour ratio of 2.5[64] (where a kiloRayleigh (kR) is 10$^{13}$ ph s$^{-1}$ m$^{-2}$ column radiating into 4π sr); determination of the planet centre pixel by fitting computed limb and terminator profiles, and subtraction of reflected sunlight from the planet disc, modelled by fitting Minnaert functions. The STIS Multi-Anode Microchannel Array (MAMA) detector has a plate scale of 0.025" pix$^{-1}$ (c.f. 0.06" pix$^{-1}$ for NIRCam), so for comparison with the NIRCam images, we have convolved the STIS images with a 2D boxcar filter of width 3 × 3 pixels. We then converted the H$_2$ emission intensity in kR into W pix$^{-1}$ assuming a representative 121 nm wavelength, and then projected the images to produce energy flux $E_{H_2}$ in W m$^{-2}$ on the planet in a similar fashion to that described above for the NIRCam images. As is usual practice, the FUV emission is projected to a spheroid 240 km above the 1-bar level. For the IRF analysis, the FUV data were reduced as above but extracted with a 3 s integration time to facilitate direct comparison with NIRCam values.

## Projections parallel to statistical ovals and keograms

Following previous studies of the FUV auroras e.g. ref. 61, we have transformed the images into a coordinate system with dimensions parallel and perpendicular to the Nichols et al.[42] reference main oval. The spacing along the oval is in equal steps of auroral longitude, which is phase around a point defined by the centroid of the main emission, increasing westward from the direction oriented toward the prime meridian[65]. We have then interpolated the image along a line perpendicular to the auroral oval at each point to produce a two-dimensional image with auroral longitude as the horizontal coordinate and the locally perpendicular direction as the vertical coordinate. We use this projection to produce a keogram as we now describe.

A keogram is a method of visualising the dynamics of auroral emission, by plotting the auroral intensity along a line versus time. The result is a 2D image, with one spatial dimension and one time dimension. The spatial dimension is chosen based on the particular

analysis, and can be e.g. a straight line such as a particular meridian, or a curved path such as a statistical auroral oval or a spacecraft trajectory. The 1D intensity profiles obtained from the chosen line at different times are then stacked to create the 2D image, such as shown in Supplementary Fig. 5. In this case, the spatial coordinate is computed from the rows in the projections described above along which the REAPs propagate. Specifically, we take the mean of a strip 0.3° wide centred 3° poleward of the statistical ME. To aid visualisation we have high-pass filtered the vertical axis to highlight the rapid (~10 s) variation by subtracting a version filtered with a Gaussian kernel with standard deviation of 30 s. Vertical slices thus indicate the REAP light curve at a particular auroral longitude, as shown on the right. We have obtained the period of this pulsation from the slice shown, computed using the Scipy periodogram algorithm with a Hann window[35], as shown in Supplementary Fig. 6. Note the significant peak at 1.6 min, which also appears with undiminished prominence in the unfiltered light curve.

## Data availability

The raw HST and JWST data are available at the Mikulski Archive for Space Telescopes (MAST) archive via doi:10.17909/chb9-zp32. The reduced JWST and HST data generated in this study (for example reduced unprojected and projected NIRCam images) have been deposited in a Figshare database[66] at https://doi.org/10.25392/leicester.data.c.7428289.v3. Further data generated in this study are provided in the Source Data file. The datasets generated during and/or analysed during the current study are also available from the corresponding author upon request. Source data are provided with this paper.

## Code availability

Computer code is available on Figshare[66]. The JWST calwebb reduction pipeline code[54] and the h3ppy code[52] are available on Zenodo.

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

## Acknowledgements

This research is based on observations made with the NASA/ESA/CSA James Webb Space Telescope and the NASA/ESA Hubble Space Telescope. The data were obtained from the Mikulski Archive for Space Telescopes at the Space Telescope Science Institute, which is operated by the Association of Universities for Research in Astronomy, Inc., under NASA contract NAS 5-03127 for JWST and NASA contract NAS 5-26555 for HST. These observations are associated with HST programme 17471 and JWST programme 4566. This work was supported by STFC Grant ST/W00089X/1 (J.D.N., O.R.T.K., L.N.F. and T.K.Y.), STFC James Webb Fellowship ST/W001527/2 (H.M.), STScI programme number JWST-GO-03665.002-A provided through a grant from the STScI under NASA contract NAS5-03127 (L.M.), and JSPS KAKENHI 20KK0074 (C.T.). J.D.N. thanks Hajime Kita for kindly providing the digital Watanabe et al. data for our use in the Discussion. This research used the ALICE High Performance Computing facility at the University of Leicester.

## Author contributions

J.D.N. led the data acquisition, analysis, and paper writing. O.R.T.K. led the analysis of the NIRSpec data. J.T.C., I.d.P., L.N.F., H.M., L.M. and C.T. contributed to data acquisition and analysis. T.K.Y. contributed to data analysis. All authors contributed to editing the paper.

## Competing interests

The authors declare no competing interests.
