## [Transparent Peer Review file · Nature Communications]

Dynamic infrared aurora on Jupiter

Corresponding Author: Professor Jonathan Nichols

Version 0:

Reviewer comments:

Reviewer #4

(Remarks to the Author)

In reviewing this paper, I have focused on the primary task that was given to me, rather than fully reviewing as I would as a primary referee. I can declare that the authors, in my opinion, have adequately responded to the referee criticisms.

This paper clearly, in my view, satisfies the Nature Communications definition of “high quality research of interest to specialists within each field.” While perhaps the additional information now provided by the authors was needed, I do not understand the negativity that comes out from the referee remarks. This is not a Nature Magazine paper, which has (as I understand it) different requirements and needs.

By enhancing the focus on “timing” and time scales in the revised version, the authors have clearly demonstrated the significance of their work. In particular, cooling rates have been a substantial focus of previous work and this paper adds substantially to that topic. Whether or not those new rates can be tied unambiguously to a new evaluation of electron densities is irrelevant, in my view, to the suitability of this work for Nature Communications. Similarly, whether or not the authors can uniquely identify a specific wave mode for explaining the newly observed dynamic phenomena is similarly irrelevant. The authors have done their job in both areas, in my view, by providing hypotheses and possibilities.

Just as important in my view regarding the suitability of this article for Nature Communications, JWST is an important new asset for studying planetary systems that the authors have brought to bear on an important target that has the complexity to test the capabilities of that new asset. Other planetary scientist will, in my opinion, be greatly interested in these findings.

Electron densities associated with auroral forcing have recently been challenged by the results from Hodges et al. (2020; below). The authors may (or may not) want to comment. But again, the findings about timing are important irrespective of whether or not the authors can uniquely tie it to auroral electron densities. See: Hodges, A., Steffes, P., Bellotti, A., Waite, J. H., Brown, S., Oyafuso, F., et al. (2020). Observations and electron density retrievals of Jupiter's discrete auroral arcs using the Juno Microwave Radiometer. *Journal of Geophysical Research: Planets*, 125, e2019JE006293. <https://doi.org/10.1029/2019JE006293>.

I may have missed it but is there a reference for the claim in the latter portion of the following sentence: “As thermal emission, the H+3 volume emission rate depends on both the H 3 density and its temperature, though overall more strongly on the former than the latter.

The paper is full of “run-on” sentences that make reading difficult. I strongly advise that many of the author’s sentences be split into two or more sentences.

Have all of the different features addressed been discussed the literature? Some literature citation is strongly needed particularly for the “DAR” since it appears not to be a prominent feature in the UV images.

(Remarks on code availability)

Not applicable.

Reviewer #5

(Remarks to the Author)

As a replacement referee for one no longer serving, I have been charged with determining whether the authors' current response to the original referees' concerns has been adequately addressed.

As mentioned in the introductory paragraph of the rebuttal text, the revisions in response to the referees' comments have been extensive, involving the removal of some material and including clarifications and augmentations. Discussion of the morphology of the auroras was reduced in order to concentrate instead on the temporal variability of the NIR H3+ aurora in comparison to the quite different FUV aurora, which was observed simultaneously with the JWST observations. The augmentation compares the first JWST images of Jupiter's NIR H3+ aurora to concurrent HST images of its FUV H2 aurorae. This comparison illuminates the lifetime of H3+ in response to auroral precipitation, revealing less radiative cooling of the aurorae than previously thought (as H3+ is supposed to be the dominant cooling agent in outer planet upper atmospheres and their aurora). This result is important because it impacts the energy balance of Jupiter's auroral atmosphere and may bear on the question - open since the Voyager epoch - of "What is the source of the excess heating of Jupiter's upper atmosphere?" (i.e., the "energy crisis"). The augmentation also argues for the scientific importance of the JWST imaging with ~3 sec exposures in characterizing the short-term time scale variations of Jupiter's H3+ emission in contrast to the non-simultaneous scans used to construct "images" strip-by-strip from JUNO's rotational scans over a significantly longer observation time of 10-15 min at 30 sec per spin. Since measurement of the H3+ number densities is a key scientific goal of the extended JUNO mission, it is scientifically important at JWST as well - more so due to the JWST data's being true images (as opposed to JUNO's rotational scans) so that any temporal changes in the H3+ morphology between exposures several seconds apart can be resolved. This enables the novel analyses shown in the new Fig. 6, which constrains the lifetime of Jupiter's auroral H3+ in the NAR flare. The augmentation further includes an analysis of accompanying JWST NIRSpec data to support the stated conclusion that the bright DAR emission is due to elevated H3+ column density, rather than elevated temperature (thus constraining future modeling to understand this brightest feature of the Jovian aurora).

The main concerns of referee #1 for the original version are significance, speculation, informing beyond existing knowledge, auroral electron densities, and the source of the REAPs. These were resolved in the revised version either by removing material, concentrating on temporal variability and comparison with the FUV, clarification, and convincing rebuttal arguments, as detailed in the rebuttal text.

The main concerns of referee #2; namely, a light analysis, exploration of implications, and broad importance are satisfactorily addressed in the revision by significantly augmenting the analysis regarding the energy budget, and adding a new analysis of the comparison with the FUV that constrains the H3+ lifetime, as summarized in more detail in the rebuttal text. In the process, a new section was created that contained two new Figures.

The concerns of referee #3 are the better spatial resolution of JUNO JIRAM images and overstatement of the magnitude of the time resolution improvement, and previous publication of timescales of 60 secs ("so variation times of ~10 seconds are not so surprising"). The rebuttal to the statement that the JUNO images are better, aside from the statement that that claim was never made in the manuscript, is that these are not images at all, being assembled from strips scanned across a small portion of the aurora. So the JUNO data risk smearing features that vary on timescales shorter than the 10-15 min of the JIRAM scans. By contrast, the JWST exposures were ~3 sec (not ~10 sec) and they were true images that covered the field simultaneously over the exposure. More is given in the rebuttal text.

In summary, the revised version of the manuscript satisfactorily addresses the issues raised by the three referees for the original version. There should be interest in the results and images reported here from the magnetospheric and auroral science communities, atmospheric scientists, astronomers searching for such bright emissions brown dwarfs and exoplanets, and the general science community sharing an interest in planetary physics, aurorae and JWST results.

Typos found: In the Eq.(2) integrand, "x" should be replaced by "t".
Three lines later, "tau_" should be replaced by "tau_ H3+ "

Laurence Trafton

(Remarks on code availability)

Reviewer #6

(Remarks to the Author)

The paper reports interesting results obtained with a the new JWST/NIRCam instrument. Despite the work is very well presented and discussed, I think the novelty of the presented data is not so striking that deserves to be published in Nature Communications. To me this paper should be published in other scientific journals specialised in auroral observations. Just the time resolution of the images does not add a strong improvement in what is already known about Jupiter auroras

(Remarks on code availability)

We thank the reviewers for their thoughtful consideration of our manuscript and the previous revisions. We have copied below in black text the relevant parts of the reviewers' reports that included the remaining suggestions for minor revision or concerns. Our response to each point is in blue, describing the modifications we have made to the manuscript.

Reviewer #4

Electron densities associated with auroral forcing have recently been challenged by the results from Hodges et al. (2020; below). The authors may (or may not) want to comment. But again, the findings about timing are important irrespective of whether or not the authors can uniquely tie it to auroral electron densities. See: Hodges, A., Steffes, P., Bellotti, A., Waite, J. H., Brown, S., Oyafuso, F., et al. (2020). Observations and electron density retrievals of Jupiter's discrete auroral arcs using the Juno Microwave Radiometer. *Journal of Geophysical Research: Planets*, 125, e2019JE006293. <https://doi.org/10.1029/2019JE006293>.

We thank the reviewer for highlighting the Hodges et al. study. It is indeed relevant, although as the reviewer suggests, the electron density values determined in that study are significantly different to other works. We have added the following statement to the Discussion: "It is worth also comparing our results with the values in the auroral regions of up to 10^{10} cm^{-3} , determined by Hodges et al. [38] from Juno microwave radiometer measurements. Those values are clearly significantly different, though may represent upper limits to local density values. Comparison of future JWST data with e.g. Juno radio occultation measurements will further constrain the electron number densities."

I may have missed it but is there a reference for the claim in the latter portion of the following sentence: "As thermal emission, the H₃⁺ volume emission rate depends on both the H₃⁺ density and its temperature, though overall more strongly on the former than the latter."

This sentence perhaps needed clarifying, and we have changed it to: "As thermal emission, the H₃⁺ volume emission rate depends on both the H₃⁺ density and its temperature, though in the jovian auroral region, changes to the density tend to dominate the observed radiance variation [Johnson et al., 2017]"

Johnson, R.E., H. Melin, T.S. Stallard, C. Tao, J.D. Nichols, and M.N. Chowdhury. 'Mapping H₃⁺ + Temperatures in Jupiter's Northern Auroral Ionosphere Using VLT-CRIRES'. *Journal of Geophysical Research: Space Physics*, 10 July 2018. <https://doi.org/10.1029/2018JA025511>.

The paper is full of "run-on" sentences that make reading difficult. I strongly advise that many of the author's sentences be split into two or more sentences.

We have reviewed the text and shortened many sentences that comprised more than three clauses. Most now have two. Changes have been made throughout the manuscript.

Have all of the different features addressed been discussed the literature? Some literature citation is strongly needed particularly for the “DAR” since it appears not to be a prominent feature in the UV images.

Many of the broad morphological features (main emission, swirl, active regions, footprints etc.) have been observed in the FUV, and for brevity we have cited the review of Grodent et al. (2015) [ref. 32], which discusses all these features. The REAPs and IFP pulsations are new, and as such don't appear in previous literature. However, the DAR region is more complex – multiple patches and arcs have been observed in this region in the FUV, and we now include some discussion of this as below, along with an augmented discussion and comparison with previous Juno observations of poleward H3+ forms. Text added to the Discussion:

“The region poleward of the main emission on the dusk side often exhibits bright, variable patches and arcs in the FUV (Bonfond et al. 2016, Nichols et al. 2017). In the NIR, relatively bright patches have been observed from ground-based observations on the dusk side (Baron et al., 1991, Johnson et al., 2017a,b), though with lower spatial resolution than in the images presented here. It is unclear whether these are related to the DAR emission, which is clearly an isolated form separated from the main emission. As discussed above, Juno JIRAM has observed forms (arcs and patches) poleward of or branching off the main emission (Dinelli et al. 2019, Gérard et al. 2018, 2023), and the H2 and H3+ intensities have been compared. The poleward forms observed by Gérard et al. (2023) were present in regions of broadly elevated FUV intensity, even if the locations of the FUV and H3+ maxima did not precisely coincide. Note also those forms were observed in pseudo-images built over 10-15 min. In contrast, the images obtained here show the bright DAR emission to be a highly variable form, separated from the main emission and in a region of generally low FUV intensity.”

Reviewer #5

Typos found: In the Eq.(2) integrand, “x” should be replaced by “ t’ “.

Three lines later, “tau_” should be replaced by “ tau_ H3+ “

We thank the referee for highlighting these typos, which we have now corrected.

Please find Reviewer #6's comments below in black, and our response in blue.

Reviewer #6

I think the novelty of the presented data is not so striking that deserves to be published in Nature Communications. To me this paper should be published in other scientific journals specialised in auroral observations. Just the time resolution of the images does not add a strong improvement in what is already known about Jupiter auroras.

We respectfully disagree with Reviewer #6's assertion that our new data and analysis improves "just the time resolution of the images" of auroral observations. We present the first JWST images of Jupiter's infrared H_3^+ auroras; true images that properly record the morphology of these emissions with exquisite sensitivity and at an unprecedented 3 s resolution. These are landmark observations that contrast with Juno JIRAM's "pseudo-images" built up over 10-15 mins and ground-based observations with typically similar integration times (we discuss the one exception with 45 to 110 s integrations in the paper). The revealed variability illustrates that the H_3^+ concentration varies much more rapidly than has been previously inferred, and we identify several auroral phenomena that have never been seen before. Furthermore, we have obtained simultaneous true images of the far-ultraviolet H_2 auroras with HST; again, this has never been done before. This enables us to compare the FUV and NIR light curves and determine the H_3^+ response to auroral energy input. This is particularly important since H_3^+ is thought to be an important part of the planet's upper atmosphere energy budget. Using our combined data, we have determined the H_3^+ lifetime during an auroral flare by fitting the convolution of an impulse response function to the FUV emission; this analysis has not been previously possible with any other data set. It allowed us to estimate the electron density in Jupiter's auroral ionosphere; a major goal of the Juno radio occultation experiments to be undertaken later this year and next. We have also determined that H_3^+ does not balance the auroral gas heating in regions of transient/bursty auroras. Further, we have discovered that the brightest and most variable H_3^+ feature does not have an FUV counterpart. This cannot be easily explained, given current understanding of the production of H_3^+ and H_2 aurora and the jovian high latitude electron population. Many searches for such emissions from exoplanets and brown dwarfs have their foundations in our understanding of their generation at Jupiter; our results highlight a fundamental gap in our understanding that needs to be addressed. Far from improving "just the time resolution of the images", all these various discoveries have significant implications for our understanding of the upper atmospheres of the gas giants, exoplanets and brown dwarfs, and are thus of broad interest.